# Evaluation of qualitative and quantitative data of Y-90 imaging in SPECT/CT and PET/CT phantom studies

**Agata Kubik**[1]*, **Anna Budzyńska**[1,2], **Krzysztof Kacperski**[1,3], **Maciej Maciak**[4], **Michał Kuć**[4], **Piotr Piasecki**[5], **Maciej Wiliński**[4], **Marcin Konior**[6], **Mirosław Dziuk**[1,2], **Edward Iller**[6]

1 Department of Nuclear Medicine, Military Institute of Medicine, Warsaw, Poland, 2 Affidea Mazovian PET/CT Medical Centre, Warsaw, Poland, 3 Particle Acceleration Physics and Technology Division (TJ1), National Centre for Nuclear Research, Otwock—Świerk, Poland, 4 Radiological Metrology and Biomedical Physics Division (H2), National Centre for Nuclear Research, Otwock—Świerk, Poland, 5 Department of Interventional Radiology, Military Institute of Medicine, Warsaw, Poland, 6 National Centre for Nuclear Research, Radioisotope Centre POLATOM, Otwock—Świerk, Poland

* akubik@wim.mil.pl

## Abstract

### Introduction

We aimed to assess the feasibility of SPECT and PET Y-90 imaging, and to compare these modalities by visualizing hot and cold foci in phantoms for varying isotope concentrations.

### Materials and methods

The data was acquired from the Jaszczak and NEMA phantoms. In the Jaszczak phantom Y-90 concentrations of 0.1 MBq/ml and 0.2 MBq/ml were used, while higher concentrations, up to 1.0 MBq/ml, were simulated by acquisition time extension with respect to the standard clinical protocol of 30 sec/projection for SPECT and 30 min/bed position for PET imaging. For NEMA phantom, the hot foci had concentrations of about 4 MB/ml and the background 0.1 or 0.0 MBq/ml. All of the acquired data was analysed both qualitatively and quantitatively. Qualitative assessment was conducted by six observers asked to identify the number of visible cold or hot foci. Inter-observer agreement was assessed. Quantitative analysis included calculations of contrast and contrast-to-noise ratio (CNR), and comparisons with the qualitative results.

### Results

For SPECT data up to two cold foci were discernible, while for PET four foci were visible. We have shown that CNR (with Rose criterion) is a good measure of foci visibility for both modalities. We also found good concordance of qualitative results for the Jaszczak phantom studies between the observers (corresponding Krippendorf's alpha coefficients of 0.76 to 0.84).

In the NEMA phantom without background activity all foci were visible in SPECT/CT images. With isotope in the background, 5 of 6 spheres were discernible (CNR of 3.0 for the

**Data Availability Statement:** Data included in Supporting Information files.

**Funding:** KK received no specific funding for this work. All of the remaining authors were funded by

project OPUS-13 no 2017/25/B/ST7/01745 funded by National Science Centre of Poland (https://www.ncn.gov.pl). The funders had no role in study design, data collection and analysis, decision to publish, or preparation of the manuscript.

smallest foci). For PET studies all hot spheres were visible, regardless of the background activity.

## Conclusions

PET Y-90 imaging provided better results than Bremsstrahlung based SPECT imaging. This indicates that PET/CT might become the method of choice in Y-90 post radioembolization imaging for visualisation of both necrotic and hot lesions in the liver.

## Introduction

Selective Internal Radiation Therapy (SIRT) using Y-90 microspheres, also known as radioembolization procedure, is an emerging method of liver cancer treatment performed in leading nuclear medicine facilities. It involves the administration of active microspheres to close proximity of the tumour through hepatic arteries, which enables more direct interaction with the diseased tissue and reduces the irradiation of healthy tissue, limiting therapy side effects. Radioembolization utilises high energy β⁻ radiation emitted in Y-90 decay to destroy cancer cells [1].

Post therapeutic SPECT/CT (Y-90 Bremsstrahlung) imaging of patients who underwent radioembolization is performed mainly to confirm a proper distribution of the Y-90 microspheres within liver tumours and in order to find potential Y-90 extrahepatic leak. Due to SPECT/CT imaging limitations only visual assessment is reliable. SPECT/CT has not been used to evaluate tumour's absorbed dose to predict radioembolization results. Utilising the post-treatment PET/CT imaging for quantitative dosimetry analysis may give better treatment results enabling recognition of liver tumours with sufficient Y-90 microspheres deposition and calculation of healthy liver tissue irradiation. That information may be crucial to select patients for further locoregional liver tumours treatment and to avoid adverse events related to liver dysfunctions [2–4].

β⁻ radiation emitted by Y-90 cannot be imaged directly with nuclear medicine techniques. However, secondary radiation generated in Y-90 decay process may be registered and used for imaging purposes.

Y-90 SPECT imaging is based on registering of Bremsstrahlung photons generated as a result of in tissue deceleration of electrons originating from β⁻ decay. Since imaging of a continuous photon energy spectrum is not common in nuclear medicine imaging it results in an increase of registered artefacts and poorer image quality [5]. One of the disadvantages of Y-90 Bremsstrahlung imaging is low spatial resolution, up to 15 mm, which depends on the energy window width, collimator choice, as well as image processing [6, 7]. This originates from its inherent technical limitation of imaging a continuous radiation spectrum, which has no pronounced photopeak, and cannot be distinguished from the in-patient scatter, resulting in only a coarse representation of the microsphere biodistribution [8]. An additional challenging issue related to Y-90 Bremsstrahlung SPECT is the attenuation correction since it depends not only on the density of objects through which the photon passes, but also on the photon's energy [5]. Y-90 Bremsstrahlung scintigraphy is considered too inaccurate for dose-response analysis, despite the possible compensation techniques for attenuation, scatter, and collimator detector response [9–11]. As an alternative Y-90 can be imaged with PET by detecting annihilation photons generated after internal pair creation in the E0 transition between the 1.76 MeV level and the ground level of the Zr-90 nucleus at the Y-90 decay. However, the branching ratio

related to the internal pair production during Y-90 decay is only $3.26 \times 10^{-5}$ pairs/decay [12]. Despite this poor ratio, the latest generation PET scanners with increased detection sensitivity due to the use of Time of Flight (TOF) technology enable the registration of the annihilation photons, thus acquiring high resolution Y-90 PET images of microspheres biodistribution. This requires only small adjustments to scan technique, such as acquisition time and number of bed positions, as well as fine-tuning of image reconstruction parameters [8, 13]. Moreover, Y-90 PET images have been proved to be suitable for quantification and thus for potential use for post SIRT dosimetry [14, 15].

Y-90 PET reflects the tumour heterogeneity better than traditional Bremsstrahlung Y-90 SPECT, as referred to in literature [13]. This is crucial in reliable estimation of Y-90 microspheres' activity distribution, which is essential for the calculation of the dose delivered to the hepatic tumours.

In order to extend our previous work with NEMA phantom filled with Y-90 resin microspheres, which was an aspect of our study [16], we aimed to further assess the feasibility of SPECT and PET Y-90 imaging, and to compare these modalities in phantom imaging. We aimed at visualising both hot and cold foci in phantoms for different concentrations of yttrium-90 chloride. We imaged both hot and cold regions, because in real clinical setting, due to hepatic tumours' heterogeneity, not only active tumours ('hot' foci) are observed, but also ones with a necrotic core ('cold' focus) surrounded by active margins ('hot' regions) [2, 17–19].

Visual assessment of medical images consisting of marking of discernible foci or the intensity of radiopharmaceutical's uptake is a very subjective method. Therefore, one of the aims of our study was to assess the inter-observer agreement concerning the number of visible foci.

Furthermore, we wanted to provide a quantitative parameter, which would enable us to determine which of the imaged regions were distinguishable from the background. We aimed to use quantitative measures of the image contrast and noise to objectively assess focus visibility in hybrid SPECT/CT and PET/CT data.

## Materials and methods

Data was acquired by scanning of two phantoms: the *Pro-SPECT Performance* Jaszczak phantom and *Pro-NM NEMA NU2* NEMA phantom, over two imaging sessions.

### Jaszczak phantom studies

The imaging sessions were designed to provide data with varying concentration of the Y-90. For the Jaszczak phantom we used Y-90 concentrations of 0.1 and 0.2 MBq/ml. In order to avoid excessive exposure, higher concentrations of isotope in the phantom, up to 1.0 MBq/ml, were simulated by extending the acquisition time with respect to the clinical standard protocols for Y-90 SPECT and PET imaging. We have conducted four SPECT and four PET acquisitions. In PET imaging some of the simulated isotope concentrations were obtained by reconstructing smaller temporal portions of the longer acquisitions using list mode data. The particular parameters of each data acquisition are presented in Table 1.

### NEMA phantom studies

In the NEMA phantom the hot foci had Y-90 concentration of about 4.0 MBq/ml while the background had 0.0 or 0.1 MBq/ml (Table 1). These concentrations were dictated by our previous experience as published [16]. The standard acquisition protocol was used for both SPECT and PET phantom imaging.

**Table 1. SPECT and PET imaging set-up for Jaszczak and NEMA phantoms.**

| | Imaging session number | Y-90 concentration in the Jaszczak phantom [MBq/ml] | | SPECT OR PET ACQUISITION PROTOCOL | Y-90 CONCENTRATION IN THE NEMA PHANTOM (at the beginning of acquisition) [MBq/ml] | | SPECT OR PET ACQUISITION PROTOCOL |
|---|---|---|---|---|---|---|---|
| | | JASZCZAK PHANTOM IMAGING | | | NEMA PHANTOM IMAGING | | |
| | | Real | Simulated | | Background | Hot foci | |
| SPECT | I. | 0.10 | - | Standard (30 sec/proj.) | 0 | 4.04 | Standard (30 sec/proj.) |
| | I. | 0.10 | 0.20 | Extended (60 sec/proj.) | | | |
| | II. | 0.20 | - | Standard (30 sec/proj.) | 0.10 | 3.83 | Standard (30 sec/proj.) |
| | II. | 0.20 | 1.00 | Extended (150 sec/proj.) | | | |
| PET | I. | 0.1 | - | Standard (30 min/bed) | 0 | 3.99 | Standard (30 min/bed) |
| | I. | 0.1 | 0.2 | **Reconstructed** List Mode data to 30 min/bed **from 60 min/bed** | | | |
| | I. | 0.1 | 0.2 | Extended (60 min/bed) | | | |
| | II. | 0.20 | - | Standard (30 min/bed) | 0.10 | 3.87 | Standard (30 min/bed) |
| | II. | 0.20 | 0.20 | **Reconstructed** List Mode data to 30, 60, 90 and 120 [min/bed] **from 150 min/bed** | | | |
| | | | 0.40 | | | | |
| | | | 0.60 | | | | |
| | | | 0.80 | | | | |
| | II. | 0.20 | 1.00 | Extended (150 min/bed) | | | |

## PET/CT imaging protocol

PET/CT imaging was performed with a hybrid PET/CT system (Discovery 710, GE Health-care). First, a scout view and a low-dose 64-slice CT scan was performed for attenuation correction of PET emission data and localisation of the phantom structures. CT scan was acquired with a tube voltage of 140 kV in the helical mode with current modulation in the range of 40–120 mA. The rotation speed was1.25 s$^{-1}$, helical thickness 3.75 mm, standard reconstruction, slice thickness 1.25 mm, matrix size 512x512.

Following CT, three-dimensional PET images were acquired. Emission scan time was 30 min per bed position (15.7 cm with 23% bed overlap). The number of bed positions was two for the Jaszczak and one for the NEMA phantom. Emission data was corrected for geometrical response, detector efficiency, system dead time, random coincidences, scatter and attenuation.

For non-attenuation corrected images the 3D iterative reconstruction technique (GE VUE Point HD) with 2 iterations/24 subsets and a filter cut-off of 6.4 mm was used. The matrix size was 192x192. Attenuation corrected images were obtained with the use of 3D-OSEM iterative reconstruction method. It was conducted with TOF PET reconstruction algorithm (GE VUE Point FX) and a resolution recovery algorithm (GE SharpIR) with 4 iterations/32 subsets and a filter cut-off of 3.0 mm. The matrix size was 256x256.

## SPECT/CT imaging protocol

SPECT/CT imaging was performed with a hybrid dual head gamma camera (Infinia VCHWK4, GE Healthcare). The energy window for Y-90 Bremsstrahlung SPECT imaging was 140 keV ± 100% and HEGP collimators were used. For each scan 60 projections were acquired in step & shoot mode, with the angular step of 6$^{\mathrm{O}}$. Total angular range was 360$^{\mathrm{O}}$

(180$^O$ per detector). Body-contour orbit was used to keep the camera close to the phantom during the entire SPECT acquisition. The acquisition time was set for 30 seconds per projection. The matrix size was 128x128 with a pixel size of 4.42 mm. OSEM reconstruction with 2 iterations and 15 subsets was used. Butterworth filter with a cut-off frequency of 0.5 cycles/cm and a power of 10 was used as a 3D postfilter.

Following emission tomography, a CT scan was performed in the axial mode with the tube voltage of 140 kV and the current of 5 mA. The X-ray tube velocity was 2.6 revolutions per minute. The matrix size was 512x512.

## Attenuation correction in SPECT images

Due to the fact that Y-90 SPECT imaging is based on registration of Bremsstrahlung with a continuous energy spectrum, wide energy window (140 keV ± 100%) was used. The automatically created attenuation correction maps assumed the emission of monoenergetic photons of 140 keV, so they could not be assumed to provide correct results. Therefore, we have employed an empirically chosen effective attenuation map to provide better attenuation correction for our data. Similar approach is frequently used in cases when exact correction is not possible or practical, e.g. in the absence of scatter correction or multienergy isotopes [11, 20, 21]. To determine the effective attenuation coefficients images of the phantom were reconstructed using attenuation correction maps generated from the CT scan for 140 keV, rescaled by several constant factors. For each image the coefficient of variation (COV) has been computed for a large ROI in a uniform cross-section of the phantom. The images were smoothed by a low pass filter in order to reduce the statistical noise, so that COV reflects mainly the non-uniformity due to imperfect attenuation correction. The scaling factor 0.6 for which the COV was the smallest has been chosen to obtain the effective attenuation map used throughout the study. This choice has also been confirmed by assessment of the profiles through the uniform section of the phantom (Fig 1).

## Data analysis

Data collected in the study was analysed both qualitatively and quantitatively.

Qualitative assessment was conducted by human observers, who were asked to identify the number of cold (in the Jaszczak Phantom), or hot (in the NEMA Phantom) foci. To blind the observers and ensure that all of the images were encoded so that the dataset's name would not allow for identification of the imaging modality, acquisition parameters and the concentration of isotope during the assessment, all datasets were renamed with randomized numbers. Nineteen datasets were prepared containing both SPECT and PET acquisitions along with corresponding CT images. Four of those were studies of NEMA Phantom, and the remaining 15 of the Jaszczak phantom.

Six members of the research team assessed the images visually. This group included two physicians experienced in the analysis of imaging data in nuclear medicine, two medical physicists working in the nuclear medicine field and two engineers with no previous experience of assessing such data. To provide equal conditions for all analysts, all persons involved in qualitative assessment worked in the same room, using the same dual-monitor workstations (Xeleris 4.1 XFL). All participants reviewed the data using the same applications with identical set up options. To avoid differences, all images were opened by an experienced workstation user.

Each observer could adjust the viewing parameters individually, by choosing the zoom and windowing, depending on their preferences. They could view all slices of the phantom, with no time restraints.

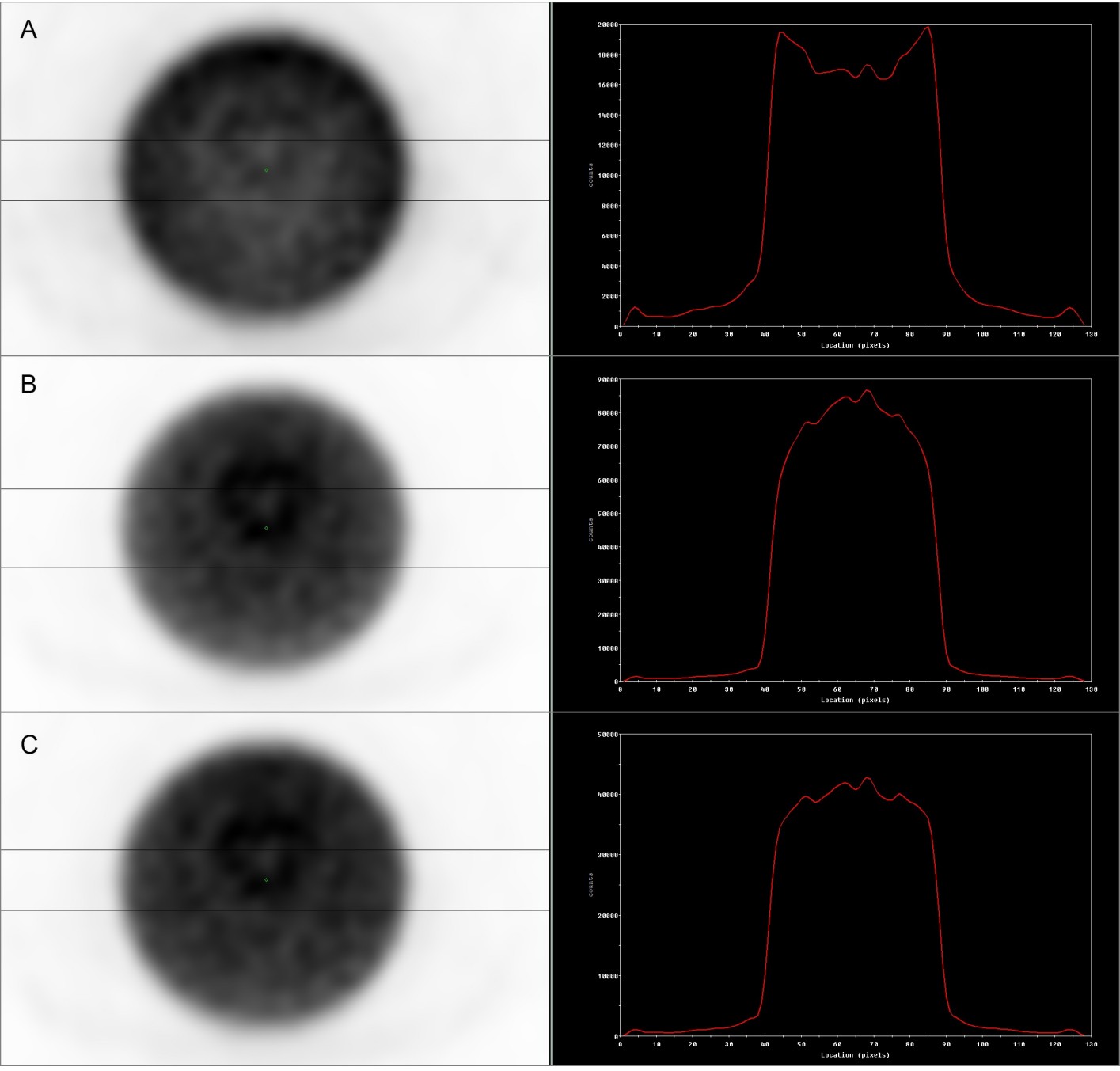

**Fig 1. Jaszczak phantom SPECT data.** Images obtained by summing 5 transaxial slices of phantom section with uniform concentration of Y-90 (on the left) and corresponding profiles through the centre if each image (on the right). **A**: non-attenuation corrected image (the shape of the profile is the result of attenuation in the phantom); **B**: image reconstructed with the original attenuation correction map (the shape of the profile indicates overcorrection of attenuation); **C**: image reconstructed with the adjusted attenuation correction map (visible improvement in the uniformity of the profile).

All images were assessed twice. At first, only SPECT and PET images were considered. First 15 datasets contained images of the Jaszczak Phantom, and the remaining four—the NEMA Phantom. The observers analysed the images in the same order and recorded their remarks on previously prepared forms. After assessing all of the datasets they moved on to the second stage, where

the same images were analysed again with the addition of CT series. Both separate and fused images were available. As in the previous stage, the observers could adjust the viewing parameters. They could not compare the scores with those given in stage one. Based on all 6 individual scores, a common accumulated number of visible foci has been determined for imaging series.

Quantitative analysis was aimed at implementation of a parameter that would differentiate visible foci from the ones that could not be discerned. Both foci to background contrast and noise in the image have a great impact on lesion detection. Therefore, the contrast-to-noise ratio (CNR) is a quantifiable parameter that should provide the information needed. The Rose criterion states that for an object to be detectable in the image, its CNR should be over a certain threshold, usually between 3 and 5 [22].

In all analysed images we have defined the cold and hot spheres based on CT images. All six regions of interest (ROIs) where then transferred onto the corresponding SPECT and PET data in order to calculate the quantitative parameters.

For cold foci in the Jaszczak phantom in SPECT data we have used the following equations for our calculations [23]:

$$CNR = \frac{C}{RMSN} = \frac{\frac{S_b - S_{mROI}}{S_b} \cdot 100\%}{\frac{\sigma_b}{S_b} \cdot 100\%} = \frac{S_b - S_{mROI}}{\sigma_b} \tag{1}$$

where C is contrast of the cold sphere, $S_b$—mean background signal, $S_{mROI}$—minimal signal in the analysed ROI, RMSN—root mean square noise and $\sigma_b$—standard deviation in the background.

For the PET images of cold foci we have implemented the following calculation methods [24]:

$$CNR = C_{ROI} \cdot \sqrt{n_{ROI}} \cdot \frac{S_b}{\sigma_b} = \frac{S_{ROI} - S_b}{S_b} \cdot 100\% \cdot \sqrt{n_{ROI}} \cdot \frac{S_b}{\sigma_b} = \frac{S_{ROI} - S_b}{\sigma_b} \cdot \sqrt{n_{ROI}} \cdot 100\% \tag{2}$$

where $C_{ROI}$ is the sphere to background contrast, $n_{ROI}$- the number of pixels in the ROI, $S_{ROI}$- the mean signal in the ROI.

For hot spheres in the NEMA phantom we have modified the equation for contrast to [22]:

$$C = \frac{S_{ROI} - S_b}{S_{ROI}} \cdot 100\% \tag{3}$$

Where $S_{ROI}$ is the mean signal in the hot sphere. For images with Y-90 activity in the background the RMSN was used as the measure of noise in the data, while in the case of images obtained without any activity in the background, noise was measured as the mean signal in the background.

Due to noise in the PET images all calculations were conducted after applying the Wiener filter (PSF = 5, noise to signal ratio = 0.11) [25].

## Inter-observer agreement

For the Jaszczak phantom studies inter-observer agreement assessment was performed for the qualitative analysis results. Agreement between different observers was appraised. That included comparisons between not only all six analysts, but also between the three groups of researchers (engineers, medical physicists and physicians) and inside each of these groups (that is the agreement between the two engineers, two physicists and two physicians). The values of Krippendorff's alpha coefficients were calculated for each emission tomography modality (PET and SPECT) both with and without the corresponding CT scans.

We have assumed the following interpretation of the Krippendorff's alpha coefficient values: α ≥ 0.800 –very good concordance; 0.667 ≤ α < 0.800 –acceptable agreement; α < 0.667 –unacceptable concordance [26].

No agreement analysis was performed for the NEMA phantom qualitative assessment, as only two SPECT/CT and two PET/CT acquisitions were performed (one with and one without background activity for each modality).

## Results

### Jaszczak phantom

In the qualitative analysis of SPECT images of the Jaszczak phantom no more than two cold foci have been marked as visible with the diameter of the smallest visible sphere of 25.4 mm (Fig 2). On the other hand, up to four spheres were visible on PET images with the smallest diameter of visible foci of 15.9 mm (Fig 3). The full data regarding sphere discernibility is presented in Table 2.

Calculated CNR values were in good agreement with qualitative data. Based on Rose criterion we have chosen 4 as the border value for foci imaged in SPECT, as no regions with CNR lower than 3.7 were deemed visible, and no spheres with CNR higher than 4.3 were assessed as indiscernible (Fig 4A). For PET data we have set the border value of CNR at 3, since no ROIs in which CNR was lower than 2.5 represented spheres marked as visible, and all of those with CNR above 3.5 were deemed discernible (Fig 4B).

### NEMA

In the NEMA phantom with no background activity all foci were clearly visible in SPECT/CT images. With isotope in the background, 5 of 6 spheres were discernible for all of the observers (Fig 5). For PET studies all hot spheres were visible, regardless of the background activity (Fig

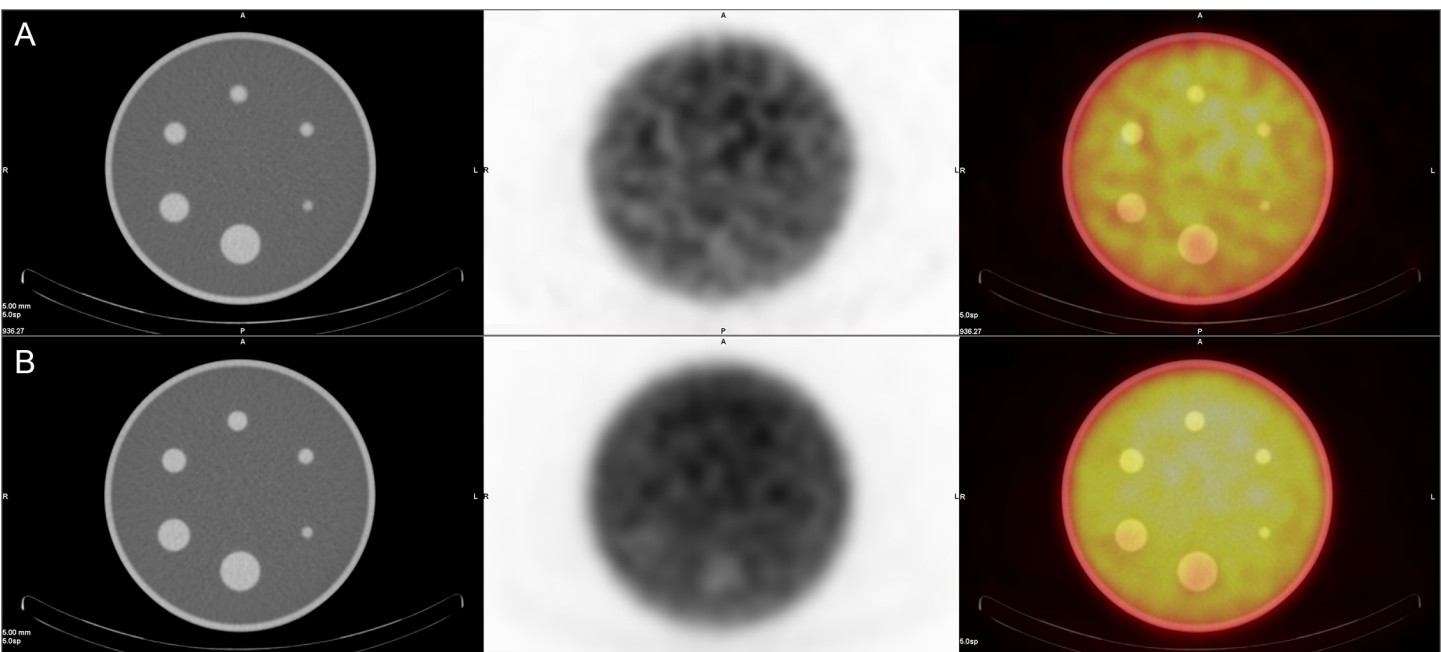

**Fig 2. SPECT/CT images of the Jaszczak phantom with different isotope concentrations. A**: Y-90 concentration of 0.1 MBq/ml, **B**: Y-90 simulated concentration of 1 MBq/ml. From left to right CT image, SPECT image and fusion of the two.

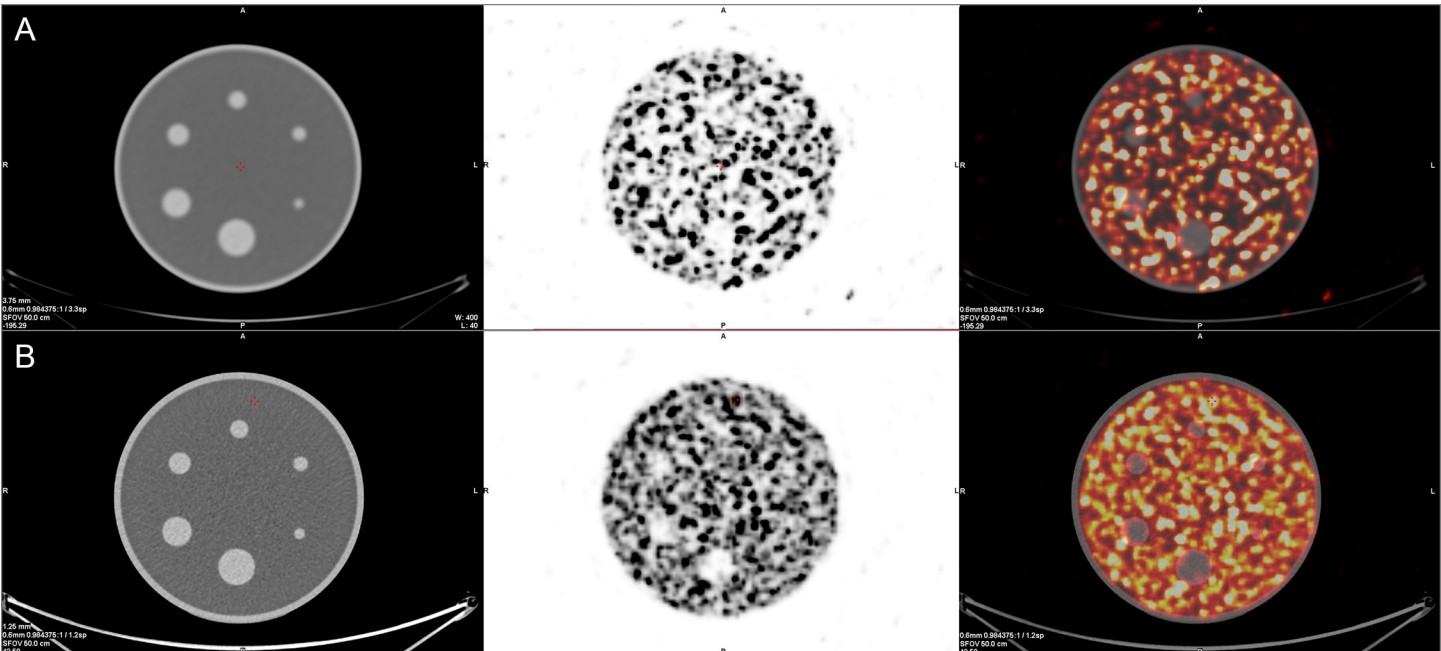

**Fig 3. PET/CT images of the Jaszczak phantom with different isotope concentrations. A**: Y-90 concentration of 0.1 MBq/ml, **B**: Y-90 simulated concentration of 1 MBq/ml. From left to right CT image, PET image and fusion of the two.

6). These observations were confirmed by high values of CNR. For SPECT data the smallest hot foci in the image with background Y-90 activity had the smallest CNR of 3, while all others exceeded 4 (Fig 7A). For PET images the calculated CNR was well above the requirement of discernibility based on Rose criterion (Fig 7B).

## Inter-observer agreement

We also found good concordance of qualitative results for the Jaszczak phantom studies between the observers. The values of concordance indicator (Krippendorff's alpha) inside each

**Table 2. Number of visible cold foci in all analysed Jaszczak phantom images.**

| MODALITY | IMAGE NUMBER | Concentration of Y-90 (* denotes simulated concentration) [MBq/ml] | NUMBER OF VISIBLE FOCI (IN QUALITATIVE ASSESSMENT) |
|---|---|---|---|
| SPECT | 1 | 0.2* | 1 |
| SPECT | 2 | 0.2 | 1 |
| SPECT | 3 | 0.1 | 1 |
| SPECT | 4 | 1.0* | 2 |
| PET | 1 | 0.1 | 1 |
| PET | 2 | 0.1 | 1 |
| PET | 3 | 0.2* | 4 |
| PET | 4 | 0.2 | 3 |
| PET | 5 | 0.2 | 2 |
| PET | 6 | 0.4* | 3 |
| PET | 7 | 0.6* | 3 |
| PET | 8 | 0.8* | 4 |
| PET | 9 | 1.0* | 4 |

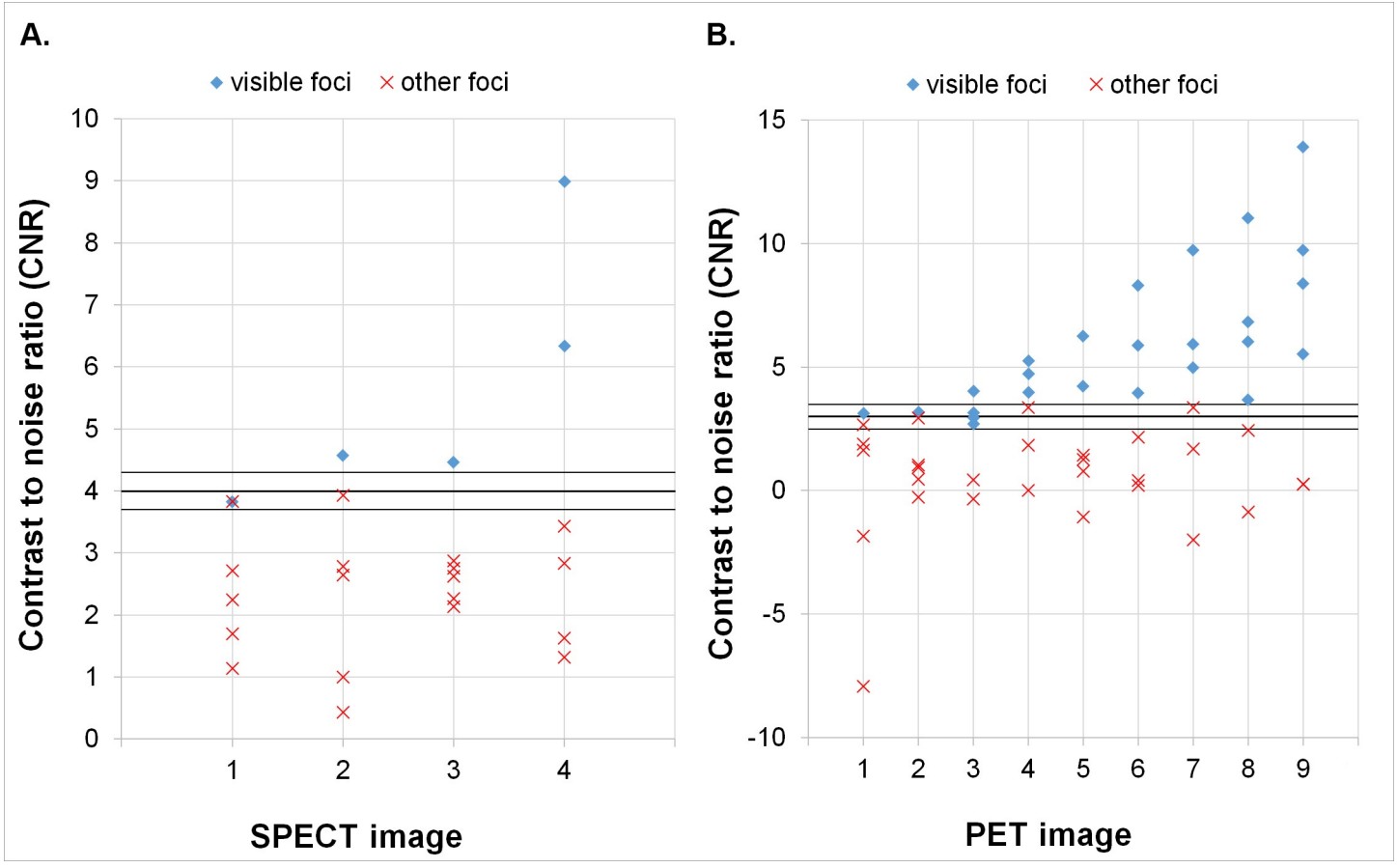

**Fig 4.** CNR values of all cold foci in SPECT (A) and PET (B) imaging. Images numbered as in Table 2. In **A** the lines depict the border values of visibility of the region based on Rose criterion (3.7; **4**; 4.3); in **B** the lines depict the border values of visibility of the region based on Rose criterion (2.5; **3**; 3.5).

of the three groups of observers varied from 0.73 to 0.92. It is worth noting that for images analysed with the aid of CT scans the observed agreement was very good (Krippendorff's alpha ≥ 0.8). The lowest, but still highly acceptable concordance was noted for the comparison of all six observers' assessments. The agreement between physicians, physicists and engineers was also very good, as indicated by Krippendorff's alpha values of 0.84 and 0.79 for analysis of the emission data without and with the assistance of CT images respectively. Full data is presented in Table 3.

## Discussion

In our study we have shown good agreement between qualitative and quantitative assessment of foci visibility in both NEMA and Jaszczak Phantom. Comparison of the quantitative parameters with the averaged qualitative assessment was possible due to the very good concordance demonstrated between all of the observers, as well as between and inside each of the considered groups.

Comparison of calculated parameters and marks awarded by the observers showed that in SPECT images for cold foci in the Jaszczak phantom the CNR value of 4 defined the distinguishability of the lesions. Similarly for PET images of this phantom we have found that CNR value of 3 differentiated between distinguishable and indistinguishable foci. However, it is

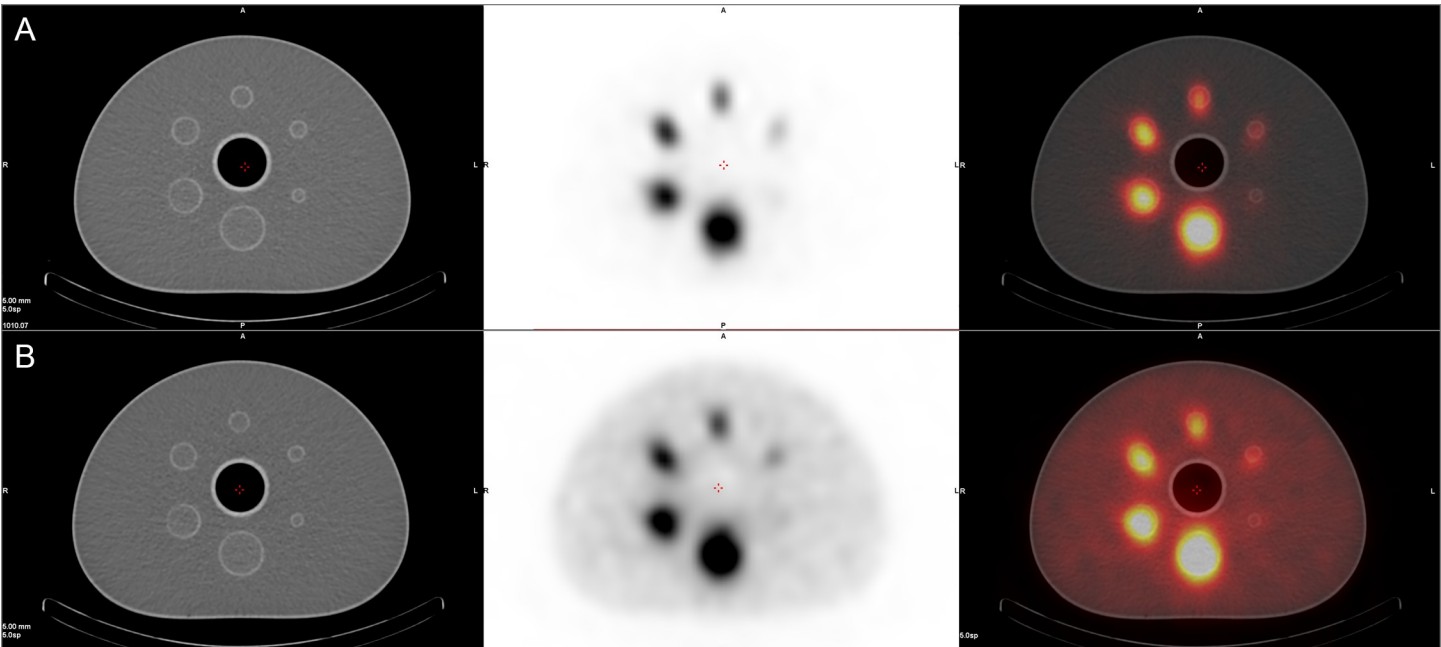

**Fig 5.** SPECT/CT images of the NEMA phantom without (**A**) and with Y-90 in the background (**B**). From left to right CT image, SPECT image and fusion of the two.

important to notice that CNR values close to 4 (for SPECT) and 3 (for PET) indicate border-line possibility of detection. Both of those values are concordant with the general rule for object detectability based on Rose criterion.

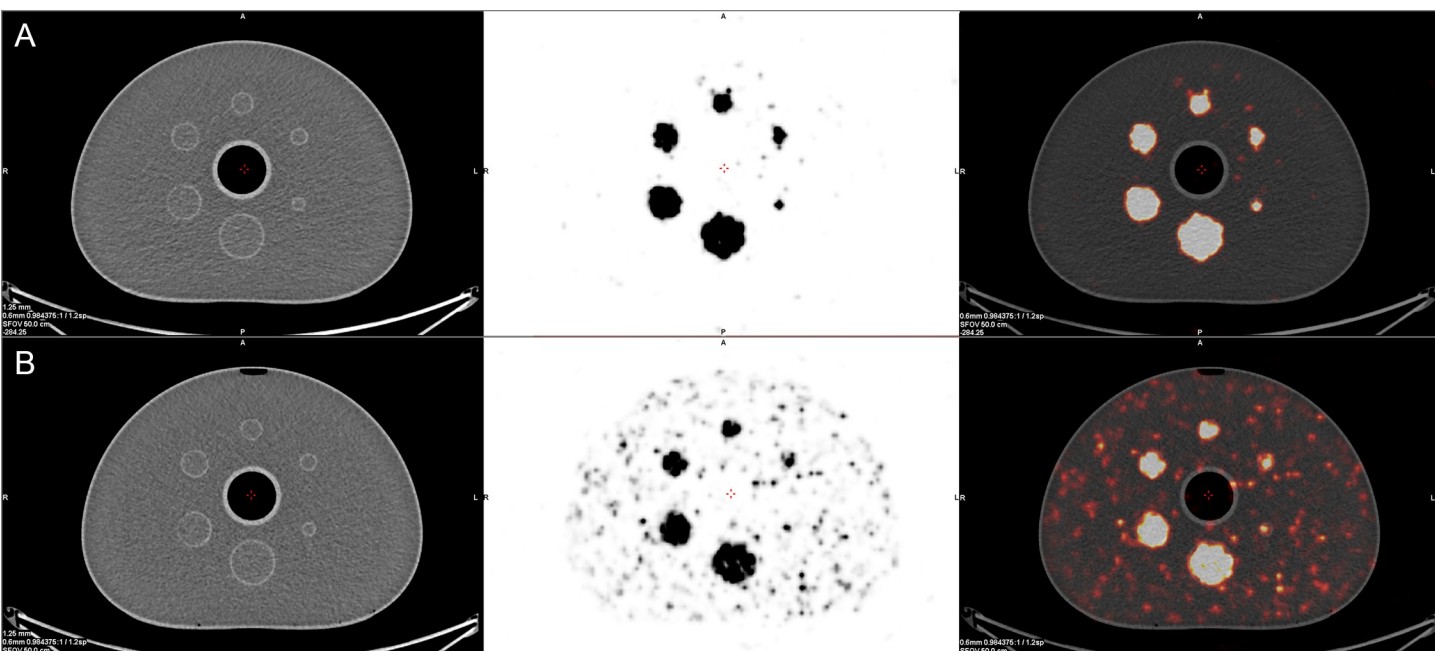

**Fig 6.** PET/CT images of the NEMA phantom without (**A**) and with Y-90 in the background (**B**). From left to right CT image, PET image and fusion of the two.

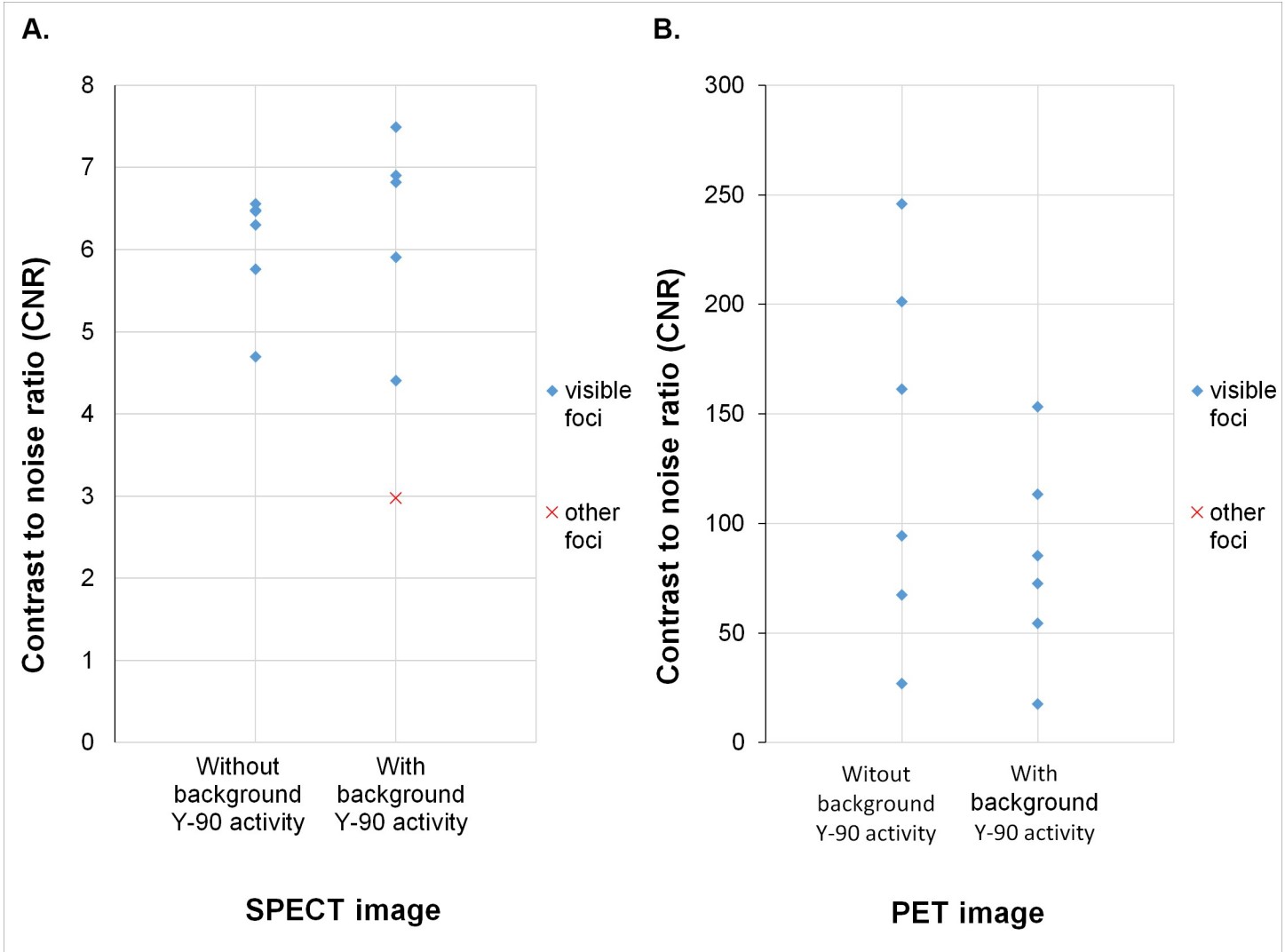

**Fig 7.** CNR values of hot foci in the NEMA phantom calculated from SPECT (**A**) and PET (**B**) images. In **A** the red marker for the image with Y-90 activity in the background represents the smallest sphere (CNR = 3). All but one foci were clearly visible, as confirmed by CNR values exceeding the threshold of 4 set by the Rose criterion. In **B** all foci were clearly visible, as confirmed by CNR values exceeding the threshold of 5 set by the Rose criterion.

**Table 3. Inter-observer agreement analysis results for Jaszczak phantom studies.**

| NUMBER OF OBSERVERS | DESCRIPTION | KRIPPENDORFF'S ALPHA COEFFICIENT | |
|---|---|---|---|
| | | Emission data only (SPECT and PET images) | SPECT and PET images with corresponding CT |
| 2 | engineers | 0.90 | 0.91 |
| 2 | medical physicists | 0.73 | 0.86 |
| 2 | physicians | 0.92 | 0.80 |
| 6 | all analysts | 0.77 | 0.76 |
| 3 | inter-group comparison (engineers vs physicists vs physicians) | 0.84 | 0.79 |

For NEMA phantom data we have found that all of the spheres had CNR values which conformed with the detectability rules defined in the Jaszczak phantom data analysis. As expected, in PET data we have noted that CNR values obtained from the images with background Y-90 activity were lower than those with cold background (Fig 7B). However it was not the case in SPECT images. The calculated CNR values for data with background activity proved higher for the biggest spheres and lower for the smallest ones when compared with the data acquired with cold background (Fig 7A). This may be explained by the increased contrast in the latter with simultaneous decrease in signal to noise ratio.

Our SPECT/CT images appear smooth and diffuse, which poses difficulties with distinguishing of the smallest hot foci. This aspect of Y-90 Bremsstrahlung imaging is also noted by other authors [8]. As a result, in patient studies assessment of activity within sub-centimetre tumours or tumour vascular thrombosis is often suboptimal.

SPECT imaging based on registration of Bremsstrahlung in a wide energy window is appropriate for qualitative analysis, for example for the visual assessment of lesions in post-therapeutic imaging. However, it poses some issues when quantitative analysis is concerned, where reliable and precise attenuation correction is needed. Automatically generated attenuation correction maps, which are accurate for the exact photon energy of 140 keV, proved insufficient for our purposes (Fig 1B). In our work we have empirically optimised attenuation correction maps, which enabled us to achieve a more reliable and uniform image of activity distribution in the phantom (Fig 1C). Further work is planned to optimise our acquisition protocols in order to improve attenuation correction, including, but not limited to, registration of Bremsstrahlung in multiple energy windows [27, 28].

In SPECT data from Jaszczak phantom no more than two cold spheres were detectible. It is however important to note that even though they were marked as visible, the calculated parameters indicated that all of those foci, apart from the ones with the highest isotope concentration in the background, were on the verge of not being seen. On the other hand, in PET images up to 4 foci were clearly distinguishable. Therefore it suggests that SPECT imaging might not be the optimal method for imaging of Y-90 activity distribution, especially considering the heterogeneity of often imaged hepatic lesions [13].

PET Y-90 imaging provided better results than Bremsstrahlung based SPECT imaging. This indicates that PET/CT might become the method of choice in Y-90 post-radioembolization imaging for visualisation of both necrotic lesions and hot lesions in the liver, as it represents a technological leap from the traditional SPECT/CT imaging. The better quality of PET Y-90 imaging was also described by other authors [8, 13].

However, it is worth noting that while PET scanning and reconstruction protocol has been thoroughly validated for quantitative imaging, the processing of SPECT data, based on Bremsstrahlung emission, remains much more challenging. For PET, the very low positron emission probability is the main fundamental limit to image quality. On the other hand, in the processing of SPECT data there is still much room for improvement, e.g. by precise modelling of the generation and propagation of Bremsstrahlung in the reconstruction algorithm and using all the spectral data available.

## Supporting information

**S1 File. Data supporting calculations.** The S1 File includes numerical data used in quantitative analysis of SPECT and PET images in separate data sheets labelled '**SPECT**' and '**PET**' respectively. The next data sheet labelled '**QualitativeAssessment**' includes results from observers. The last data sheet ('**Attenuation Correction-COV**') presents calculated COV values. (XLSX)

## Author Contributions

**Conceptualization:** Agata Kubik, Anna Budzyńska.

**Data curation:** Agata Kubik, Anna Budzyńska.

**Formal analysis:** Agata Kubik, Anna Budzyńska, Krzysztof Kacperski.

**Investigation:** Agata Kubik, Anna Budzyńska, Maciej Maciak, Michał Kuć, Piotr Piasecki, Maciej Wiliński, Marcin Konior, Mirosław Dziuk.

**Methodology:** Agata Kubik, Anna Budzyńska, Maciej Maciak, Piotr Piasecki.

**Project administration:** Edward Iller.

**Resources:** Mirosław Dziuk, Edward Iller.

**Software:** Agata Kubik, Michał Kuć.

**Supervision:** Mirosław Dziuk, Edward Iller.

**Visualization:** Agata Kubik, Anna Budzyńska.

**Writing – original draft:** Agata Kubik, Anna Budzyńska.

**Writing – review & editing:** Agata Kubik, Anna Budzyńska, Krzysztof Kacperski, Maciej Maciak, Piotr Piasecki, Edward Iller.

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
