## [Decision Letter · Decision Letter 0]

18 Nov 2020

PONE-D-20-30873

Evaluation of qualitative and quantitative data of Y-90 imaging in SPECT/CT and PET/CT phantom studies

PLOS ONE

Dear Dr. Kubik,

Thank you for submitting your manuscript to PLOS ONE. After careful consideration, we feel that it has merit but does not fully meet PLOS ONE’s publication criteria as it currently stands. Therefore, we invite you to submit a revised version of the manuscript that addresses the points raised during the review process.

We look forward to receiving your revised manuscript.

Kind regards,

Pradeep K. Garg, Ph.D.

Academic Editor

PLOS ONE

Journal Requirements:

2. Please include captions for your Supporting Information files at the end of your manuscript, and update any in-text citations to match accordingly. Please see our Supporting Information guidelines for more information: http://journals.plos.org/plosone/s/supporting-information

Reviewers' comments:

Reviewer's Responses to Questions

**Comments to the Author**

1. Is the manuscript technically sound, and do the data support the conclusions?

Reviewer #1: Yes

2. Has the statistical analysis been performed appropriately and rigorously? 

Reviewer #1: Yes

3. Have the authors made all data underlying the findings in their manuscript fully available?

Reviewer #1: Yes

4. Is the manuscript presented in an intelligible fashion and written in standard English?

Reviewer #1: Yes

5. Review Comments to the Author

Reviewer #1: This is an interesting and useful manuscript describing phantom experiments to investigate Y-90 imaging. These results will positively impact the nuclear medicine community at large. With some suggestions to improve readability, I recommend publication.

The most significant issue with the manuscript it is too lengthy. While I applaud the authors for including the level of detail present in the manuscript, I feel it makes the manuscript unwieldy and lessens the overall impact. Removing some minor details (suggestions in itemized comments below) and reworking the figures/tables to be more user friendly are general suggestions.

Detailed comments:

• Fig 1 is not needed. The referenced phantoms are well-known and unless there are specific modifications to the phantoms used in this study, I would recommend removal of Fig 1.

• It is unclear how many acquisitions of each type were performed. From tables 1-4, it appears that some of the simulated data was reconstructed as a smaller temporal portion of one long acquisition. If this is correct, please clarify.

• Tables 1, 2, 3, and 4 could be combined or reworked? As currently designed, they cost almost 2 pages of text. I suspect this information could be provided to the reader in paragraph form.

• How were the imaging sessions chosen for the Jaszczak phantom studies?

• In the discussion of the PET/CT protocol, it sounds as if the CT parameters selected were the same as the site’s standard clinical protocol. Since these were only evaluated qualitatively, perhaps these parameters could be deleted without negatively impacting the manuscript? Similar consideration for SPECT/CT discussion.

• Attenuation correction for SPECT Images: Is there anything in the literature that would support the empirical attenuation correction method utilized in this study?

• Equations 1-3 could be simplified into one equation.

• Equation 4 and 5 could also be simplified into one equation (I believe C in Eq 5 should be CROI?).

• Jaszczak phantom results: I would suggest identifying the smallest foci that could be identified (by size) as well as the number of foci that were visible. This could be important as ultimately, the reader would want to determine a minimum detectability size based on this analysis.

• I feel Table 5 could be reworked. I realize you use Image Number in Fig 4,5, 8, and 9, but I found this to be difficult to interpret and required quite a bit of scrolling to review. Perhaps add minimum size detectability? Or consider removing and including in the text itself?

• Perhaps combine Fig 4, 5, 8, and 9 into one 4 panel figure? Fig 8 and 9 also need legends explaining the difference between blue and red diamonds (and why diamonds instead of x?). Would also be useful to have image labels be descriptors (if it can be fit) instead of numbers (especially if you remove table 5).

• Table 6: Any reason the krippendorff’s alpha went down for physicians when adding CT? That seems counter-intuitive.

• Attenuation correction in SPECT. Do you have a qualitative measure to support the improved uniformity? I think it’s difficult to assess the uniformity visually from the images provided in Figure 10.

• Line 333: Very strong statement that “SPECT imaging is not the optimal method for imaging…” While your study suggests that, I would argue you can’t quite support such a strong statement as of yet.

• Line 341: “close to optimal” I don’t know what this means exactly. Perhaps use different language to confer this point.

6. PLOS authors have the option to publish the peer review history of their article (what does this mean?). If published, this will include your full peer review and any attached files.

Reviewer #1: No

---

## [Author Response · Author response to Decision Letter 0]

16 Jan 2021

We have revised our file names and applied changes according to the guidelines for the supporting information.

Point-by-point response to the reviewers comments and concerns.

• Fig 1 is not needed. The referenced phantoms are well-known and unless there are specific modifications to the phantoms used in this study, I would recommend removal of Fig 1.

We agree. Figure 1 has been removed and all the other figures have been renumbered accordingly.

• It is unclear how many acquisitions of each type were performed. From tables 1-4, it appears that some of the simulated data was reconstructed as a smaller temporal portion of one long acquisition. If this is correct, please clarify.

Thank you for pointing this out. The text has been changed to clearly state the number of actual acquisitions and datasets reconstructed as a smaller portion of one long acquisitions.

Text added (lines 113-115):

‘We have conducted four SPECT and four PET acquisitions. In PET some of the simulated isotope concentrations were obtained by reconstructing smaller temporal portions of the longer acquisitions using list mode data.’

• Tables 1, 2, 3, and 4 could be combined or reworked? As currently designed, they cost almost 2 pages of text. I suspect this information could be provided to the reader in paragraph form.

Thank you for indicating the issue with the tables. We have reworked tables 1-4 into one table. We felt that providing all of the information regarding imaging set up directly in the text would become very muddled. We hope that the new format will be a good compromise between reducing space needed and including all of the information. Some of the information in it is also essential for integrity between the manuscript and the additional data files we provided.

• How were the imaging sessions chosen for the Jaszczak phantom studies?

We are not sure if we understand the question correctly. However we will try to answer the question as fully as possible. 

With regards to the concentrations chosen, they were dictated by our concerns about radiation safety as well as our previous studies (as mentioned with NEMA phantom imaging). The acquisition parameters were based on clinical protocols used in our institution, with changes only in those cases, when we needed to simulate higher concentrations. As for imaging the phantom in two separate sessions we needed time to allow for the activity in the phantom to go down enough to dispose of the contaminated water, before filling it anew. We also had to take clinical schedule of the Nuclear Medicine Department into consideration, as regular patient studies take precedence over research imaging.

• In the discussion of the PET/CT protocol, it sounds as if the CT parameters selected were the same as the site’s standard clinical protocol. Since these were only evaluated qualitatively, perhaps these parameters could be deleted without negatively impacting the manuscript? Similar consideration for SPECT/CT discussion.

We agree that the description of the CT imaging parameters may seem lengthy. We have included them for completeness of the imaging protocol description, as the clinical practices may differ from site to site. We also believe that for SPECT/CT these parameters are important since the CT images are used for attenuation correction. However, we have reduced the description by excluding some of the parameters in order to shorten the text.

• Attenuation correction for SPECT Images: Is there anything in the literature that would support the empirical attenuation correction method utilized in this study?

We have rephrased the appropriate passage in the text (lines 156 - 159) in order to describe our attenuation correction more precisely and added two references where a similar approach using effective attenuation coefficients is applied. 

Moreover, quantitative measurements of the coefficient of variation (COV) have been made on the reconstructed images in order to choose the rescaling factor of the attenuation map as the one which minimizes image non-uniformity. The procedure has been briefly described in lines 160 - 167.

Additionally, in order to shorten the manuscript, and since the attenuation correction method was not intended as one of our study’s aims, we have removed the subsection ‘Attenuation Correction’ from the Results section, as all of the information is now included in Materials and Methods.

• Equations 1-3 could be simplified into one equation.

We agree that these equations could be simplified into one. Since we believe it is important to underline the differences in chosen methods of contrast, noise and CNR calculations we chose to include all steps needed to arrive at the final equation.

• Equation 4 and 5 could also be simplified into one equation (I believe C in Eq 5 should be CROI?).

Thank you for pointing out the mistake- C in former equation 5 has been changed to CROI. Equations 4 and 5 have been compiled into one equation, while also including the steps for the calculations.

• Jaszczak phantom results: I would suggest identifying the smallest foci that could be identified (by size) as well as the number of foci that were visible. This could be important as ultimately, the reader would want to determine a minimum detectability size based on this analysis.

Information about the relevant foci size has been added: in SPECT images 25.4 mm and in PET 15.9 mm (lines 241 – 243).

• I feel Table 5 could be reworked. I realize you use Image Number in Fig 4,5, 8, and 9, but I found this to be difficult to interpret and required quite a bit of scrolling to review. Perhaps add minimum size detectability? Or consider removing and including in the text itself?

Thank you for this remark. However, we feel that in the final version of the paper when the table and images are placed side by side or at least one under the other it would be easier to consult a nearby table than try to locate the relevant information in the text. We hope that with figures 4 and 5 being reworked into one image, and 8 and 9 into one image it might require less scrolling. 

• Perhaps combine Fig 4, 5, 8, and 9 into one 4 panel figure? Fig 8 and 9 also need legends explaining the difference between blue and red diamonds (and why diamonds instead of x?). Would also be useful to have image labels be descriptors (if it can be fit) instead of numbers (especially if you remove table 5).

In order to maintain uniform labels in all of the images we decided to keep the numbers. They denote the images in order of increasing number of visible foci, and unfortunately, with the many PET series, it became illegible when we tried describing the series accurately and in detail with descriptors. Because of our decision to keep table 5 (now accordingly renumbered), the numbers are still explained in the manuscript. We hope that in the final version of the article the table and image will be placed close enough to consider both at the same time.

Explanation of colours of the markers has been added to the description. As well as the needed legends. The red diamonds have been corrected to be x, in order to maintain consistency throughout the manuscript.

• Table 6: Any reason the krippendorff’s alpha went down for physicians when adding CT? That seems counter-intuitive.

While it does seem counter intuitive we believe that it might be connected to two different approaches of the physicians. One of them used CT to exclude false positives, while the other added more foci. We believe that such approach might be due to the preference of either risking false positives or false negatives in clinical practice.

• Attenuation correction in SPECT. Do you have a qualitative measure to support the improved uniformity? I think it’s difficult to assess the uniformity visually from the images provided in Figure 10.

 We have calculated COV for uniform sections of the phantom and added the explanation about it in Materials and Methods section as described previously.

• Line 333: Very strong statement that “SPECT imaging is not the optimal method for imaging…” While your study suggests that, I would argue you can’t quite support such a strong statement as of yet.

We have redacted the phrase to ‘Therefore it suggests that SPECT imaging might not be the optimal method for imaging […]’ in line 323. 

• Line 341: “close to optimal” I don’t know what this means exactly. Perhaps use different language to confer this point.

The sentence has been changed to:

‘However, it is worth noting that while the PET scanning and reconstruction protocol has been thoroughly validated for quantitative imaging, the processing of SPECT data, based on bremsstrahlung emission, remain much more challenging.’ In lines 331-333.

---

## [Editor Report · Decision Letter 1]

27 Jan 2021

Evaluation of qualitative and quantitative data of Y-90 imaging in SPECT/CT and PET/CT phantom studies

PONE-D-20-30873R1

Dear Dr. Kubik,

We’re pleased to inform you that your manuscript has been judged scientifically suitable for publication and will be formally accepted for publication once it meets all outstanding technical requirements.

Kind regards,

Pradeep K. Garg, Ph.D.

Academic Editor

PLOS ONE

Additional Editor Comments (optional):

Authors satisfactorily answered all concerns raised by the reviewers earlier. The changes made within the revised manuscript adequately reflect and includes the response to those comments.